# Metastasis, characteristic, and treatment of breast cancer in young women and older women: A study from the Surveillance, Epidemiology, and End Results registration database

**Xiaokang Gao[1], Fengxia Zhang[1], Qiwang Zhou[1], Hui Xu[1], Jie Bian[2]***

1 Department of Thyroid and Breast Surgery, The Affiliated Hospital of Yangzhou University, Yangzhou University, Yangzhou, Jiangsu Province, P. R. China, 2 Department of Medical, The Affiliated Hospital of Yangzhou University, Yangzhou University, Yangzhou, Jiangsu Province, P. R. China

* bianjieer@outlook.com

## Abstract

### Background

Younger age is an independent risk factor for breast cancer (BC) prognosis, and BC in young women is often considered more aggressive. BC patients with different age and molecular subtypes have different metastasis patterns and survival. Herein, we aim to explore the metastasis patterns, characteristics and treatment methods of young patients with BC, and to compare them with older patients.

### Methods

Data of young patients (aged ≤40 years old) and older patients (aged >40 years old) with BC were extracted from the Surveillance, Epidemiology, and End Results (SEER) registration database in 2010–2019 in this retrospective cohort study. Univariate and multivariate competing risk models and proportional hazard models were used to explore the association between different metastasis patterns and treatments and BC prognoses in young and older patients. Kaplan-Meier (KM) curves were drawn to reflect the survival probability of patients with BC who have different metastasis patterns. Also, we performed subgroup analysis of different metastasis patterns to explore the association between different treatments and overall survival (OS)/cancer specific survival (CSS) in patients with BC. The evaluation index was hazard ratios (HRs) with 95% confidence intervals (CIs).

### Results

Totally, 5,984 patients died, and 92.56% of them died from BC. There were respectively 1,089 young patients and 9,105 older patients, and we found some differences of characteristics and metastasis patterns between them. After adjusting for covariates, young patients who had brain metastasis and multiple sites metastasis seemed to have high risk of both

are available in the SEER database, https://seer.
cancer.gov/data/access.html.

**Funding:** The author(s) received no specific
funding for this work.

**Competing interests:** The authors have declared
that no competing interests exist.

lower OS and CSS. Among older patients with BC, brain metastasis, liver metastasis, and multiple sites metastasis were all positively associated with both lower OS and CSS. In young and older patients, those who not receive radiotherapy or surgery, or received non-surgery combined with radiotherapy seemed to have high risk of both lower OS and CSS. Breast-conserving surgery (BCS) and surgery combined with radiotherapy were associated with higher OS and CSS in young patients, while only older patients received surgery combined with radiotherapy had higher OS and CSS. Results of subgroup analysis indicated that for patients with different metastasis patterns, developing a personalized treatment plan is necessary.

## Conclusions

Characteristics of BC between young patients and older patients were different. Clinicians should focus on different metastasis sites and choose appropriate treatments in patients with different ages, which may improve the prognoses.

## Introduction

Breast cancer (BC) as the most common malignancy over the world affects 2.1 million women each year, and causes the highest number of cancer-related morbidity and mortality among women [1, 2]. Although BC mainly occurs in postmenopausal women, there are still about 5% of new cases of young women (≤40 years of age) annually [3]. BC in young women is often considered more aggressive and requires aggressive treatment [4]. Previous studies have showed that young women with BC have worse prognoses than those over 40 years old, and that younger age is an independent risk factor for BC prognosis [5, 6].

BC usually has a better prognosis than other aggressive cancers, and however, when distant metastasis occurs, the 5-year survival rate for patients with BC can drop to 27% [7, 8]. Epidemiological surveys showed that the mortality rate of women with BC over 40 years old was decreasing year by year, while that of young patients has stopped decreasing since 2010, which may be attributed to the rapid increase in the incidence of young metastatic BC [8, 9]. In addition, BC patients with different age and molecular subtypes have different metastasis patterns and survival. Chen et al. [10] investigated the influence of age at diagnosis on metastatic BC and patients' prognosis, and found that elder BC patients were more likely to have lung metastasis, and they had the worst both overall survival and breast cancer-specific survival comparing to younger patients. A population-based observational study also showed that BC patients with different molecular subtypes had different metastasis patterns and survivorship [11].

Herein, it is necessary to explore the metastasis patterns, characteristics and treatment methods in young patients with BC, and to compare them with older patients, in order to provide some references for identifying high-risk population of metastatic BC in different ages and accurately formulating of intervention strategies.

## Methods

### Study design and participants

Data of patients with BC were extracted from the National Cancer Institute Surveillance, Epidemiology, and End Results (SEER) database in 2010–2019 in this retrospective cohort study. SEER uses data collected and maintained by the National Center for Health Statistics (NCHS),

which is from the Population Estimates Program (PEP) data of the United States Census Bureau and mortality data in the United States. SEER database routinely collects and publishes data on patient-specific and tumor-specific characteristics for about 30 percent of population in the United States. The collected information for each case includes patient demographics, primary tumor site, tumor morphology, stage at diagnosis, treatment course, follow-up for vital status, and cause of death. The database is updated annually and available for download after completion of a data user agreement free of charge: https://seer.cancer.gov/data/access. html.

A total of 32,081 patients with BC were initially included. The exclusion criteria were (1) missing information of surgery or radiotherapy, (2) diagnosed with two or more primary cancers, (3) patients at American Joint Committee on Cancer (AJCC) T0 stage, (4) male patients, (5) missing survival information. Finally, 10,194 of them were eligible. The participants were divided into two groups according to their age of diagnosis: young patient group (≤40 years old) and older patient group (>40 years old). Since the SEER is an open-access database, and all patients' information has been de-identified, ethical review of this study was exempt from our hospital, and the patients' informed consent was waived.

## Variables collection

The study extracted variables including age, race, marital status, family income, primary site of BC [central portion, lower inner quadrant (LIQ), lower outer quadrant (LOQ), upper inner quadrant (UIQ), and upper outer quadrant (UOQ)], tumor size (≤2, 2–5, >5 cm), tumor grade (I to IV stage), the AJCC T (T1 to TX) and N (N0 to NX) stage, histological type [invasive ductal carcinoma (IDC), invasive lobular carcinoma (ILC), and infiltrating ductal mixed lobular carcinoma (IDLC)] [12], subtype [hormone receptor (HR), and human epidural growth factor receptor (HER)] [13, 14], metastatic site (bone, lung, liver, brain, other distant sites, and multiple sites), chemotherapy, surgery, radiation, and combination therapy from SEER database. More details of the data collection could be found in the official documents of SEER: https://seer.cancer.gov/archive/manuals/2021/AppendixC/Coding_Guidelines_Breast_ 2021.pdf; https://seer.cancer.gov/tools/solidtumor/Breast_STM.pdf. In addition, the study outcomes were the overall survival (OS) and cancer specific survival (CSS). The follow-up ended when patients died or until the 31 December 2019.

## Statistical analysis

Normally distributed data were described by mean ± standard deviation (Mean ± SD), and t test was used for comparation between two groups. Skewed distributed data were described by median and quartiles [M (Q1, Q3)], and rank sum test was used for comparation. Classified variables were described in terms of frequency and composition ratio [n (%)], and chi-square test ($\chi^2$) was used for comparison.

We utilized the univariate competing risk models for covariates screening associated with OS, and univariate proportional hazard models for that associated with CSS. Univariate and multivariate competing risk models and proportional hazard models were used to explore the associations between different metastasis patterns and treatments and BC prognoses in young and older patients respectively. Kaplan-Meier (KM) curves were drawn to reflect the survival probability of patients with BC who have different metastasis patterns. Also, we performed subgroup analysis of different metastasis patterns to explore the associations between different treatments and OS/CSS in patients with BC. The evaluation index was hazard ratios (HRs) with 95% confidence intervals (CIs). Bilateral $P$ <0.05 was statistically significant. All statistical analyses were performed using SAS v.9.4 software (SAS Institute, Cary, North Carolina).

## Results

### Characteristics of study population

Fig 1 shows the flowchart of participants screening. We initially included 32,081 patients with BC from the SEER database. Those who without information of surgery (n = 262) or radiotherapy (n = 17814), diagnosed with two or over primary cancers (n = 3477), at AJCC T0 stage of BC (n = 83), were male (n = 151), or without information of survival (n = 100) were excluded. Finally, 10,194 of them were eligible.

Characteristics of young and older BC patients are showed in Table 1. There were respectively 1,089 young patients and 9,105 older patients with BC. After the follow-up, 5,984 (58.70%) patients died, and among them, 5,539 (92.56%) died due to BC. The average age of participants was 59.27 years old, and most of them are White (75.60%). IDC was the most common histological type and HR+/HER2- was the most common subtype among all patients with BC. Bone metastasis accounted for the largest proportion (42.12%), and followed by multiple sites (37.89%). The numbers of patients who received chemotherapy, surgery and

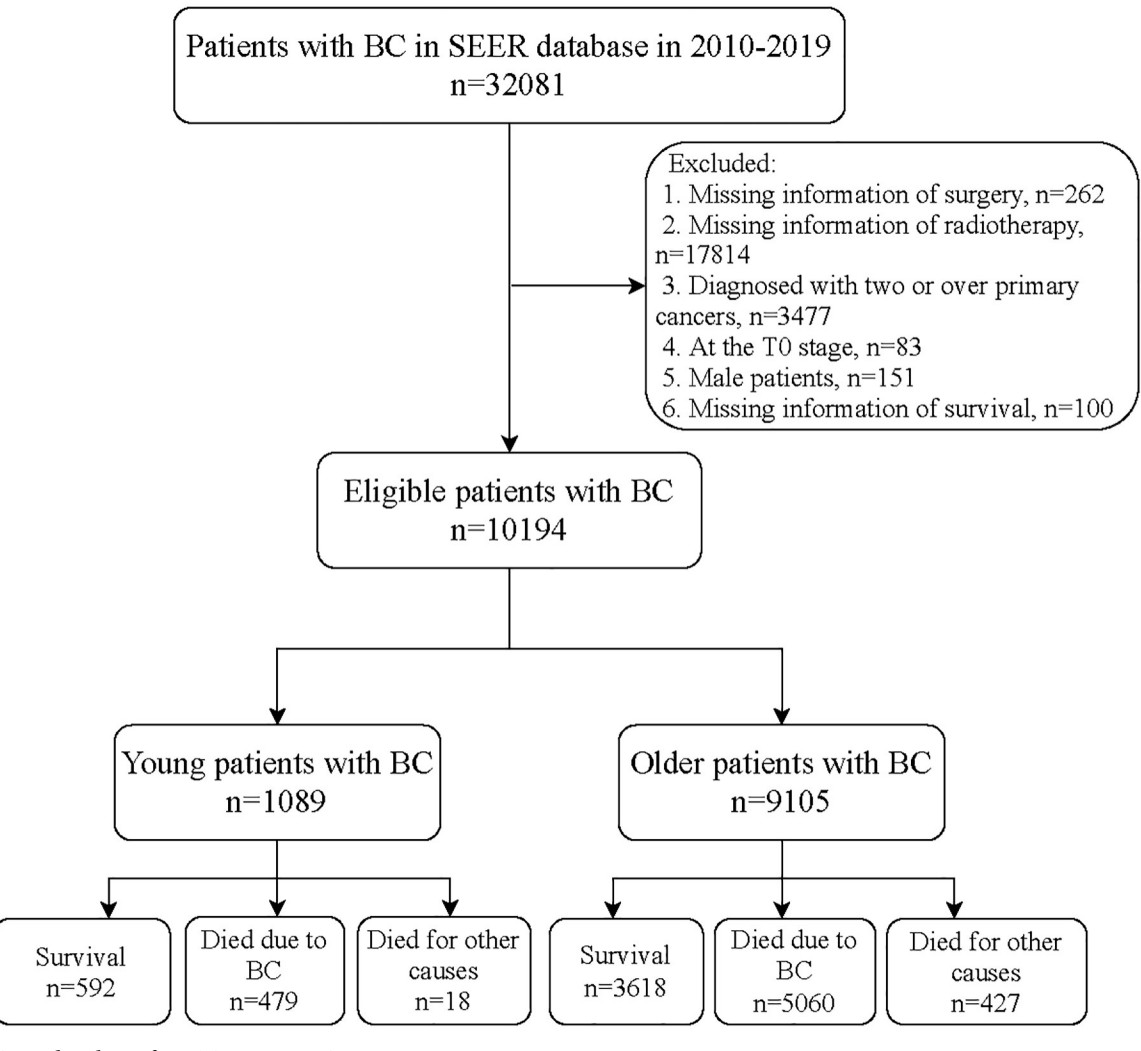

**Fig 1. Flowchart of participants screening.**

**Table 1. Characteristics of patients with BC.**

| Variables | Total (n = 10194) | Young patients (n = 1089) | Older patients (n = 9105) | Statistics | P |
|---|---|---|---|---|---|
| Age, years, Mean ± SD | 59.27 ± 14.18 | 34.96 ± 4.38 | 62.17 ± 11.99 | t = -148.89 | <0.001 |
| Race, n (%) | | | | $\chi^2$ = 33.326 | <0.001 |
| White | 7707 (75.60) | 746 (68.50) | 6961 (76.45) | | |
| Black | 1547 (15.18) | 213 (19.56) | 1334 (14.65) | | |
| Other/unknown | 940 (9.22) | 130 (11.94) | 810 (8.90) | | |
| Marital status, n (%) | | | | $\chi^2$ = 30.651 | <0.001 |
| Married (including common law) | 4609 (45.21) | 578 (53.08) | 4031 (44.27) | | |
| Unmarried (including separated, divorced, widowed, single) | 5083 (49.86) | 462 (42.42) | 4621 (50.75) | | |
| Unknown | 502 (4.92) | 49 (4.50) | 453 (4.98) | | |
| Family income, n (%) | | | | $\chi^2$ = 0.356 | 0.551 |
| < $70000 | 6496 (63.72) | 685 (62.90) | 5811 (63.82) | | |
| ≥ $70000 | 3698 (36.28) | 404 (37.10) | 3294 (36.18) | | |
| Primary site, n (%) | | | | $\chi^2$ = 23.141 | <0.001 |
| Central portion | 643 (6.31) | 37 (3.40) | 606 (6.66) | | |
| LIQ | 374 (3.67) | 48 (4.41) | 326 (3.58) | | |
| LOQ | 563 (5.52) | 72 (6.61) | 491 (5.39) | | |
| UIQ | 623 (6.11) | 70 (6.43) | 553 (6.07) | | |
| UOQ | 2513 (24.65) | 259 (23.78) | 2254 (24.76) | | |
| Unknown | 5478 (53.74) | 603 (55.37) | 4875 (53.54) | | |
| Tumor size, cm, n (%) | | | | $\chi^2$ = 2.929 | 0.403 |
| ≤ 2 | 737 (7.23) | 68 (6.24) | 669 (7.35) | | |
| 2–5 | 2334 (22.90) | 251 (23.05) | 2083 (22.88) | | |
| > 5 | 1919 (18.82) | 195 (17.91) | 1724 (18.93) | | |
| Unknown | 5204 (51.05) | 575 (52.80) | 4629 (50.84) | | |
| Tumor grade, n (%) | | | | $\chi^2$ = 35.437 | <0.001 |
| I | 484 (4.75) | 25 (2.30) | 459 (5.04) | | |
| II | 2620 (25.70) | 265 (24.33) | 2355 (25.86) | | |
| III | 3440 (33.75) | 430 (39.49) | 3010 (33.06) | | |
| IV | 61 (0.60) | 12 (1.10) | 49 (0.54) | | |
| Unknown | 3589 (35.21) | 357 (32.78) | 3232 (35.50) | | |
| AJCC T stage, n (%) | | | | $\chi^2$ = 31.989 | <0.001 |
| T1 | 594 (5.83) | 55 (5.05) | 539 (5.92) | | |
| T2 | 1820 (17.85) | 217 (19.93) | 1603 (17.61) | | |
| T3 | 989 (9.70) | 127 (11.66) | 862 (9.47) | | |
| T4 | 1842 (18.07) | 142 (13.04) | 1700 (18.67) | | |
| TX | 626 (6.14) | 53 (4.87) | 573 (6.29) | | |
| Unknown | 4323 (42.41) | 495 (45.45) | 3828 (42.04) | | |
| AJCC N stage, n (%) | | | | $\chi^2$ = 17.014 | 0.004 |
| N0 | 1193 (11.70) | 100 (9.18) | 1093 (12.00) | | |
| N1 | 2397 (23.51) | 263 (24.15) | 2134 (23.44) | | |
| N2 | 874 (8.57) | 92 (8.45) | 782 (8.59) | | |
| N3 | 964 (9.46) | 109 (10.01) | 855 (9.39) | | |
| NX | 443 (4.35) | 30 (2.75) | 413 (4.54) | | |
| Unknown | 4323 (42.41) | 495 (45.45) | 3828 (42.04) | | |
| Histological type, n (%) | | | | $\chi^2$ = 95.540 | <0.001 |
| IDC | 7298 (71.59) | 909 (83.47) | 6389 (70.17) | | |
| ILC | 774 (7.59) | 25 (2.30) | 749 (8.23) | | |

*(Continued)*

**Table 1.** (Continued)

| Variables | Total (n = 10194) | Young patients (n = 1089) | Older patients (n = 9105) | Statistics | P |
|---|---|---|---|---|---|
| IDLC | 426 (4.18) | 37 (3.40) | 389 (4.27) | | |
| Others | 1696 (16.64) | 118 (10.84) | 1578 (17.33) | | |
| Subtype, n (%) | | | | $\chi^2 = 95.874$ | <0.001 |
| HR+/HER2+ | 1606 (15.75) | 257 (23.60) | 1349 (14.82) | | |
| HR+/HER2- | 5455 (53.51) | 486 (44.63) | 4969 (54.57) | | |
| HR-/HER2+ | 867 (8.51) | 132 (12.12) | 735 (8.07) | | |
| HR-/HER2- | 1320 (12.95) | 145 (13.31) | 1175 (12.90) | | |
| Unknown | 946 (9.28) | 69 (6.34) | 877 (9.63) | | |
| Metastatic site, n (%) | | | | $\chi^2 = 58.269$ | <0.001 |
| Bone only | 4294 (42.12) | 472 (43.34) | 3822 (41.98) | | |
| Lung only | 845 (8.29) | 78 (7.16) | 767 (8.42) | | |
| Liver only | 584 (5.73) | 114 (10.47) | 470 (5.16) | | |
| Brain only | 221 (2.17) | 19 (1.74) | 202 (2.22) | | |
| Other distant sites | 387 (3.80) | 44 (4.04) | 343 (3.77) | | |
| Multiple sites | 3863 (37.89) | 362 (33.24) | 3501 (38.45) | | |
| Chemotherapy, n (%) | | | | $\chi^2 = 282.778$ | <0.001 |
| Yes | 6430 (63.08) | 940 (86.32) | 5490 (60.30) | | |
| No/unknown | 3764 (36.92) | 149 (13.68) | 3615 (39.70) | | |
| Surgery, n (%) | | | | $\chi^2 = 124.907$ | <0.001 |
| No | 4890 (47.97) | 360 (33.06) | 4530 (49.75) | | |
| BCS | 1653 (16.22) | 187 (17.17) | 1466 (16.10) | | |
| Mastectomy | 3515 (34.48) | 526 (48.30) | 2989 (32.83) | | |
| Unknown type | 136 (1.33) | 16 (1.47) | 120 (1.32) | | |
| Radiotherapy, n (%) | | | | $\chi^2 = 0.480$ | 0.488 |
| Yes | 6964 (68.31) | 754 (69.24) | 6210 (68.20) | | |
| No | 3230 (31.69) | 335 (30.76) | 2895 (31.80) | | |
| Combination therapy, n (%) | | | | $\chi^2 = 158.421$ | <0.001 |
| Non-surgery/non-radiotherapy | 333 (3.27) | 10 (0.92) | 323 (3.55) | | |
| Non-surgery/radiotherapy | 4557 (44.70) | 350 (32.14) | 4207 (46.21) | | |
| Surgery/non-radiotherapy | 2897 (28.42) | 325 (29.84) | 2572 (28.25) | | |
| Surgery/radiotherapy | 2407 (23.61) | 404 (37.10) | 2003 (22.00) | | |
| Survival, n (%) | | | | $\chi^2 = 85.821$ | <0.001 |
| Yes | 4210 (41.30) | 592 (54.36) | 3618 (39.74) | | |
| No | 5984 (58.70) | 497 (45.64) | 5487 (60.26) | | |
| Died from BC | 5539 (92.56) | 479 (96.38) | 5060 (92.22) | $\chi^2 = 11.459$ | <0.001 |
| Died from other diseases | 445 (7.44) | 18 (3.62) | 427 (7.78) | | |
| Survival time, months, M (Q$_1$, Q$_3$) | 21.00 (8.00, 42.00) | 28.00 (13.00, 50.00) | 20.00 (7.00, 41.00) | Z = 9.109 | <0.001 |

t: t test, $\chi^2$: chi-square test, Z: rank sum test.

BC: breast cancer, SD: standard deviation, LIQ: lower inner quadrant, LOQ: lower outer quadrant, UIQ: upper inner quadrant, UOQ: upper outer quadrant, AJCC: the American Joint Committee on Cancer stage, IDC: invasive ductal carcinoma, ILC: invasive lobular carcinoma, IDLC: infiltrating ductal mixed lobular carcinoma, HR: hormone receptor, HER: human epidural growth factor receptor, BCS: breast-conserving surgery, M: median, Q1: the 1st quartile, Q3: the 3rd quartile.

radiotherapy were respectively 6,430 (63.08%), 5,304 (52.03%), and 6,964 (68.31%). In addition, age, race, primary site, tumor grade, AJCC T stage, AJCC N stage, histological type, subtype, metastatic site, chemotherapy, surgery, combination therapy, and survival time were all significantly different between young patients and older patients (all $P<0.05$).

## Comparation of characteristics between young and older patients with different metastasis patterns

We compared the characteristics between young and older patients with different metastasis patterns, including bone only, liver only, lung only, brain only, other distant sites, and multiple sites (Tables 2 and 3). The results showed that age and chemotherapy were significantly different between young patients and older patients with bone, liver, lung, brain, other distant sites, or multiple sites metastases (all $P<0.05$). Between young and older patients with bone metastasis, race, marital tatus, primary site, tumor grade, AJCC T stage and N stage, histological type, subtype, surgery, combination therapy, and survival were different (all $P<0.05$). Survival was significantly different between young and older patients who had liver metastasis (all $P<0.05$). Race and survival time were different between young and older patients with lung metastasis (all $P<0.05$). Radiotherapy and combination therapy were found to be different between young patients and older patients who had other distant sites metastasis (all $P<0.05$). In addition, variables including race, tumor grade, AJCC T stage, histological type, subtype, chemotherapy, surgery, combination therapy, and survival were all significantly different in patients who had multiple sites metastasis between the two groups (all $P<0.05$).

## Association between metastasis patterns and prognosis in young and older patients with BC

We first screened the covariates associated with OS/CSS in young and older patients with BC respectively. As shown in S1 Table, race, marital status, family income, tumor size, AJCC T and N stage, histological type, subtype, metastatic site, chemotherapy, and surgery were significantly associated with both OS and CSS in young patients with BC (all $P<0.05$). In addition to the above variables, age, primary site, tumor grade, and radiotherapy were also associated with OS and CSS in older patients with BC (S2 Table).

Then, we explored the association between metastasis patterns and prognosis in young and older patients with BC respectively (Table 4). After adjusting for covariates, young patients with BC who had brain metastasis or multiple sites metastasis seemed to have high risk of both lower OS [brain metastasis: HR = 2.28, 95%CI: (1.23–4.24); multiple sites: HR = 1.72, 95%CI: (1.39–2.14)] and CSS [brain metastasis: HR = 2.61, 95%CI: (1.27–5.35); multiple sites: HR = 1.55, 95%CI: (1.24–1.93)]. Among older patients with BC, brain metastasis, liver metastasis, and multiple sites metastasis were positively associated with lower OS [brain metastasis: HR = 1.58, 95%CI: (1.33–1.87); liver metastasis: HR = 1.32, 95%CI: (1.16–1.51); multiple sites: HR = 1.86, 95%CI: (1.75–1.98)] and CSS [brain metastasis: HR = 1.67, 95%CI: (1.34–2.07); liver metastasis: HR = 1.24, 95%CI: (1.07–1.44); multiple sites: HR = 1.88, 95%CI: (1.75–2.01)]. In addition, the KM curves showed the survival probability of young and older patients with BC who have different metastasis patterns respectively (Fig 2).

## Association between different treatments and prognosis in young and older patients with BC

We also explored the relationship between different metastasis patterns and prognosis in patients with different treatments (Table 5). In young and older patients with BC, those who not receive radiotherapy or surgery, or received non-surgery combined with radiotherapy seemed to have high risk of both lower OS and CSS (all $P<0.05$), and however, breast-conserving surgery (BCS) and surgery combined with radiotherapy were associated with higher OS and CSS in young patients (all $P<0.05$). In addition, older patients who had surgery combined with radiotherapy had higher OS and CSS (all $P<0.05$).

**Table 2. Characteristics between young and older patients with bone, liver, and lung metastases.**

| Variables | Bone only | | | Liver only | | | Lung only | | |
|---|---|---|---|---|---|---|---|---|---|
| | Young patients (n = 472) | Old patients (n = 3822) | P | Young patients (n = 114) | Old patients (n = 470) | P | Young patients (n = 78) | Old patients (n = 767) | P |
| Age, years, Mean ± SD | 35.11 ± 4.20 | 62.35 ± 12.20 | <0.001 | 34.57 ± 4.26 | 58.32 ± 11.35 | <0.001 | 34.68 ± 4.96 | 64.86 ± 13.14 | <0.001 |
| Race, n (%) | | | 0.002 | | | 0.345 | | | 0.002 |
| White | 351 (74.36) | 3056 (79.96) | | 78 (68.42) | 347 (73.83) | | 41 (52.56) | 552 (71.97) | |
| Black | 65 (13.77) | 485 (12.69) | | 23 (20.18) | 69 (14.68) | | 23 (29.49) | 129 (16.82) | |
| Other/unknown | 56 (11.86) | 281 (7.35) | | 13 (11.40) | 54 (11.49) | | 14 (17.95) | 86 (11.21) | |
| Marital status, n (%) | | | <0.001 | | | 0.332 | | | 0.108 |
| Married (including common law) | 272 (57.63) | 1753 (45.87) | | 62 (54.39) | 240 (51.06) | | 41 (52.56) | 308 (40.16) | |
| Unmarried (including separated, divorced, widowed, single) | 179 (37.92) | 1888 (49.40) | | 45 (39.47) | 213 (45.32) | | 34 (43.59) | 418 (54.50) | |
| Unknown | 21 (4.45) | 181 (4.74) | | 7 (6.14) | 17 (3.62) | | 3 (3.85) | 41 (5.35) | |
| Family income, n (%) | | | 0.849 | | | 0.067 | | | 0.795 |
| < $70000 | 297 (62.92) | 2422 (63.37) | | 65 (57.02) | 311 (66.17) | | 53 (67.95) | 510 (66.49) | |
| ≥ $70000 | 175 (37.08) | 1400 (36.63) | | 49 (42.98) | 159 (33.83) | | 25 (32.05) | 257 (33.51) | |
| Primary site, n (%) | | | <0.001 | | | 0.439 | | | 0.934 |
| Central portion | 12 (2.54) | 274 (7.17) | | 4 (3.51) | 23 (4.89) | | 4 (5.13) | 58 (7.56) | |
| LIQ | 19 (4.03) | 150 (3.92) | | 8 (7.02) | 20 (4.26) | | 3 (3.85) | 28 (3.65) | |
| LOQ | 40 (8.47) | 246 (6.44) | | 4 (3.51) | 33 (7.02) | | 4 (5.13) | 49 (6.39) | |
| UIQ | 31 (6.57) | 242 (6.33) | | 10 (8.77) | 36 (7.66) | | 6 (7.69) | 49 (6.39) | |
| UOQ | 108 (22.88) | 1016 (26.58) | | 32 (28.07) | 132 (28.09) | | 21 (26.92) | 182 (23.73) | |
| Unknown | 262 (55.51) | 1894 (49.56) | | 56 (49.12) | 226 (48.09) | | 40 (51.28) | 401 (52.28) | |
| Tumor size, cm, n (%) | | | 0.112 | | | 0.098 | | | 0.310 |
| 2–5 | 129 (27.33) | 1079 (28.23) | | 24 (21.05) | 152 (32.34) | | 25 (32.05) | 185 (24.12) | |
| > 5 | 79 (16.74) | 681 (17.82) | | 27 (23.68) | 106 (22.55) | | 20 (25.64) | 267 (34.81) | |
| ≤ 2 | 30 (6.36) | 347 (9.08) | | 18 (15.79) | 53 (11.28) | | 5 (6.41) | 47 (6.13) | |
| Unknown | 234 (49.58) | 1715 (44.87) | | 45 (39.47) | 159 (33.83) | | 28 (35.90) | 268 (34.94) | |
| Tumor grade, n (%) | | | <0.001 | | | 0.363 | | | 0.151 |
| I | 16 (3.39) | 296 (7.74) | | 0 (0.00) | 14 (2.98) | | 1 (1.28) | 28 (3.65) | |
| II | 141 (29.87) | 1217 (31.84) | | 31 (27.19) | 118 (25.11) | | 10 (12.82) | 144 (18.77) | |
| III | 168 (35.59) | 1047 (27.39) | | 56 (49.12) | 228 (48.51) | | 46 (58.97) | 401 (52.28) | |
| IV | 1 (0.21) | 12 (0.31) | | 1 (0.88) | 3 (0.64) | | 4 (5.13) | 13 (1.69) | |
| Unknown | 146 (30.93) | 1250 (32.71) | | 26 (22.81) | 107 (22.77) | | 17 (21.79) | 181 (23.60) | |
| AJCC T stage, n (%) | | | <0.001 | | | 0.152 | | | 0.272 |
| T1 | 23 (4.87) | 298 (7.80) | | 16 (14.04) | 49 (10.43) | | 5 (6.41) | 31 (4.04) | |
| T2 | 119 (25.21) | 865 (22.63) | | 21 (18.42) | 130 (27.66) | | 21 (26.92) | 140 (18.25) | |
| T3 | 58 (12.29) | 404 (10.57) | | 21 (18.42) | 70 (14.89) | | 9 (11.54) | 115 (14.99) | |
| T4 | 54 (11.44) | 610 (15.96) | | 14 (12.28) | 76 (16.17) | | 17 (21.79) | 237 (30.90) | |
| TX | 18 (3.81) | 291 (7.61) | | 2 (1.75) | 14 (2.98) | | 3 (3.85) | 21 (2.74) | |
| Unknown | 200 (42.37) | 1354 (35.43) | | 40 (35.09) | 131 (27.87) | | 23 (29.49) | 223 (29.07) | |
| AJCC N stage, n (%) | | | 0.018 | | | 0.439 | | | 0.512 |
| N0 | 52 (11.02) | 567 (14.84) | | 8 (7.02) | 60 (12.77) | | 15 (19.23) | 113 (14.73) | |
| N1 | 111 (23.52) | 934 (24.44) | | 35 (30.70) | 147 (31.28) | | 25 (32.05) | 217 (28.29) | |
| N2 | 42 (8.90) | 356 (9.31) | | 14 (12.28) | 66 (14.04) | | 7 (8.97) | 92 (11.99) | |
| N3 | 52 (11.02) | 411 (10.75) | | 15 (13.16) | 57 (12.13) | | 8 (10.26) | 104 (13.56) | |
| NX | 15 (3.18) | 200 (5.23) | | 2 (1.75) | 9 (1.91) | | 0 (0.00) | 18 (2.35) | |

*(Continued)*

**Table 2.** (Continued)

| Variables | Bone only | | | Liver only | | | Lung only | | |
|---|---|---|---|---|---|---|---|---|---|
| | Young patients (n = 472) | Old patients (n = 3822) | P | Young patients (n = 114) | Old patients (n = 470) | P | Young patients (n = 78) | Old patients (n = 767) | P |
| Unknown | 200 (42.37) | 1354 (35.43) | | 40 (35.09) | 131 (27.87) | | 23 (29.49) | 223 (29.07) | |
| Histological type, n (%) | | | <0.001 | | | 0.136 | | | 0.458 |
| IDC | 401 (84.96) | 2515 (65.80) | | 102 (89.47) | 382 (81.28) | | 61 (78.21) | 569 (74.19) | |
| ILC | 15 (3.18) | 490 (12.82) | | 2 (1.75) | 22 (4.68) | | 0 (0.00) | 10 (1.30) | |
| IDLC | 21 (4.45) | 224 (5.86) | | 3 (2.63) | 26 (5.53) | | 1 (1.28) | 18 (2.35) | |
| Other | 35 (7.42) | 593 (15.52) | | 7 (6.14) | 40 (8.51) | | 16 (20.51) | 170 (22.16) | |
| Subtype, n (%) | | | <0.001 | | | 0.414 | | | 0.114 |
| HR+/HER2+ | 123 (26.06) | 471 (12.32) | | 29 (25.44) | 106 (22.55) | | 11 (14.10) | 104 (13.56) | |
| HR+/HER2- | 262 (55.51) | 2602 (68.08) | | 39 (34.21) | 155 (32.98) | | 18 (23.08) | 264 (34.42) | |
| HR-/HER2+ | 35 (7.42) | 141 (3.69) | | 28 (24.56) | 96 (20.43) | | 11 (14.10) | 87 (11.34) | |
| HR-/HER2- | 30 (6.36) | 262 (6.86) | | 14 (12.28) | 85 (18.09) | | 31 (39.74) | 214 (27.90) | |
| Unknown | 22 (4.66) | 346 (9.05) | | 4 (3.51) | 28 (5.96) | | 7 (8.97) | 98 (12.78) | |
| Chemotherapy, n (%) | | | <0.001 | | | <0.001 | | | <0.001 |
| No/unknown | 82 (17.37) | 1744 (45.63) | | 4 (3.51) | 99 (21.06) | | 3 (3.85) | 268 (34.94) | |
| Yes | 390 (82.63) | 2078 (54.37) | | 110 (96.49) | 371 (78.94) | | 75 (96.15) | 499 (65.06) | |
| Surgery, n (%) | | | <0.001 | | | 0.207 | | | 0.208 |
| No | 143 (30.30) | 1815 (47.49) | | 6 (5.26) | 41 (8.72) | | 9 (11.54) | 142 (18.51) | |
| BCS | 82 (17.37) | 684 (17.90) | | 30 (26.32) | 156 (33.19) | | 16 (20.51) | 162 (21.12) | |
| Mastectomy | 240 (50.85) | 1282 (33.54) | | 75 (65.79) | 261 (55.53) | | 53 (67.95) | 448 (58.41) | |
| Unknown type | 7 (1.48) | 41 (1.07) | | 3 (2.63) | 12 (2.55) | | 0 (0.00) | 15 (1.96) | |
| Radiotherapy, n (%) | | | 0.194 | | | 0.165 | | | 0.249 |
| No | 119 (25.21) | 1072 (28.05) | | 61 (53.51) | 285 (60.64) | | 43 (55.13) | 474 (61.80) | |
| Yes | 353 (74.79) | 2750 (71.95) | | 53 (46.49) | 185 (39.36) | | 35 (44.87) | 293 (38.20) | |
| Combination therapy, n (%) | | | <0.001 | | | 0.216 | | | 0.100 |
| Non-surgery/non-radiotherapy | 4 (0.85) | 97 (2.54) | | 1 (0.88) | 10 (2.13) | | 0 (0.00) | 34 (4.43) | |
| Non-surgery/radiotherapy | 139 (29.45) | 1718 (44.95) | | 5 (4.39) | 31 (6.60) | | 9 (11.54) | 108 (14.08) | |
| Surgery/non-radiotherapy | 115 (24.36) | 975 (25.51) | | 60 (52.63) | 275 (58.51) | | 43 (55.13) | 440 (57.37) | |
| Surgery/radiotherapy | 214 (45.34) | 1032 (27.00) | | 48 (42.11) | 154 (32.77) | | 26 (33.33) | 185 (24.12) | |
| Survival, n (%) | | | <0.001 | | | <0.001 | | | 0.159 |
| Yes | 283 (59.96) | 1733 (45.34) | | 71 (62.28) | 211 (44.89) | | 35 (44.87) | 282 (36.77) | |
| No | 189 (40.04) | 2089 (54.66) | | 43 (37.72) | 259 (55.11) | | 43 (55.13) | 485 (63.23) | |
| Died from BC | 186 (98.41) | 1889 (90.43) | 0.002 | 42 (97.67) | 235 (90.73) | 0.126 | 41 (95.35) | 433 (89.28) | 0.208 |
| Died from other diseases | 3 (1.59) | 200 (9.57) | | 1 (2.33) | 24 (9.27) | | 2 (4.65) | 52 (10.72) | |
| Survival time, months, M (Q₁, Q₃) | 33.50 (16.00, 57.50) | 28.00 (12.00, 51.00) | <0.001 | 41.50 (19.00, 64.00) | 26.00 (11.00, 55.00) | <0.001 | 27.00 (15.00, 52.00) | 20.00 (9.00, 47.00) | 0.028 |

Statistics included t test, chi-square test, and rank sum test.

SD: standard deviation, LIQ: lower inner quadrant, LOQ: lower outer quadrant, UIQ: upper inner quadrant, UOQ: upper outer quadrant, AJCC: the American Joint Committee on Cancer stage, IDC: invasive ductal carcinoma, ILC: invasive lobular carcinoma, IDLC: infiltrating ductal mixed lobular carcinoma, HR: hormone receptor, HER: human epidural growth factor receptor, BCS: breast-conserving surgery, BC: breast cancer, M: median, Q1: the 1st quartile, Q3: the 3rd quartile.

## Relationships between treatments and BC prognosis in subgroups of different metastasis patterns

In addition, the associations between treatments and BC prognosis were explored in different metastasis patterns subgroups (Table 6). After adjusting for covariates, we found that young

**Table 3. Characteristics between young and older patients with brain, other distant sites, and multiple sites metastases.**

| Variables | Brain only | | | Other distant sites | | | Multiple Sites | | |
|---|---|---|---|---|---|---|---|---|---|
| | Young patients (n = 19) | Old patients (n = 202) | P | Young patients (n = 44) | Old patients (n = 343) | P | Young patients (n = 362) | Old patients (n = 3501) | P |
| Age, years, Mean ± SD | 36.21 ± 2.59 | 61.95 ± 11.96 | <0.001 | 35.86 ± 3.42 | 61.07 ± 12.07 | <0.001 | 34.76 ± 4.67 | 62.03 ± 11.41 | <0.001 |
| Race, n (%) | | | 1.000 | | | 0.583 | | | <0.001 |
| Black | 3 (15.79) | 35 (17.33) | | 9 (20.45) | 67 (19.53) | | 90 (24.86) | 549 (15.68) | |
| Other/unknown | 1 (5.26) | 13 (6.44) | | 8 (18.18) | 44 (12.83) | | 38 (10.50) | 332 (9.48) | |
| White | 15 (78.95) | 154 (76.24) | | 27 (61.36) | 232 (67.64) | | 234 (64.64) | 2620 (74.84) | |
| Marital status, n (%) | | | 0.701 | | | 0.343 | | | 0.109 |
| Married (including common law) | 7 (36.84) | 83 (41.09) | | 22 (50.00) | 163 (47.52) | | 174 (48.07) | 1484 (42.39) | |
| Unmarried (including separated, divorced, widowed, single) | 10 (52.63) | 108 (53.47) | | 22 (50.00) | 164 (47.81) | | 172 (47.51) | 1830 (52.27) | |
| Unknown | 2 (10.53) | 11 (5.45) | | 0 (0.00) | 16 (4.66) | | 16 (4.42) | 187 (5.34) | |
| Family income, n (%) | | | 0.864 | | | 0.205 | | | 0.624 |
| < $70000 | 13 (68.42) | 142 (70.30) | | 25 (56.82) | 228 (66.47) | | 232 (64.09) | 2198 (62.78) | |
| ≥ $70000 | 6 (31.58) | 60 (29.70) | | 19 (43.18) | 115 (33.53) | | 130 (35.91) | 1303 (37.22) | |
| Primary site, n (%) | | | 0.874 | | | 0.601 | | | 0.283 |
| Central portion | 0 (0.00) | 8 (3.96) | | 1 (2.27) | 29 (8.45) | | 16 (4.42) | 214 (6.11) | |
| LIQ | 1 (5.26) | 9 (4.46) | | 1 (2.27) | 15 (4.37) | | 16 (4.42) | 104 (2.97) | |
| LOQ | 0 (0.00) | 6 (2.97) | | 3 (6.82) | 13 (3.79) | | 21 (5.80) | 144 (4.11) | |
| UIQ | 0 (0.00) | 15 (7.43) | | 2 (4.55) | 23 (6.71) | | 21 (5.80) | 188 (5.37) | |
| UOQ | 6 (31.58) | 53 (26.24) | | 13 (29.55) | 85 (24.78) | | 79 (21.82) | 786 (22.45) | |
| Unknown | 12 (63.16) | 111 (54.95) | | 24 (54.55) | 178 (51.90) | | 209 (57.73) | 2065 (58.98) | |
| Tumor size, cm, n (%) | | | 1.000 | | | 1 | | | 0.374 |
| ≤ 2 | 2 (10.53) | 21 (10.40) | | | | | 13 (3.59) | 201 (5.74) | |
| 2–5 | 4 (21.05) | 47 (23.27) | | | | | 69 (19.06) | 620 (17.71) | |
| > 5 | 4 (21.05) | 40 (19.80) | | | | | 65 (17.96) | 630 (17.99) | |
| Unknown | 9 (47.37) | 94 (46.53) | | | | | 215 (59.39) | 2050 (58.55) | |
| Tumor grade, n (%) | | | 0.658 | | | 0.171 | | | 0.004 |
| I | 1 (5.26) | 5 (2.48) | | 3 (6.82) | 6 (1.75) | | 4 (1.10) | 110 (3.14) | |
| II | 4 (21.05) | 32 (15.84) | | 3 (6.82) | 47 (13.70) | | 76 (20.99) | 797 (22.76) | |
| III | 8 (42.11) | 83 (41.09) | | 10 (22.73) | 88 (25.66) | | 142 (39.23) | 1163 (33.22) | |
| IV | 0 (0.00) | 2 (0.99) | | 0 (0.00) | 0 (0.00) | | 6 (1.66) | 19 (0.54) | |
| Unknown | 6 (31.58) | 80 (39.60) | | 28 (63.64) | 202 (58.89) | | 134 (37.02) | 1412 (40.33) | |
| AJCC T stage, n (%) | | | 0.623 | | | | | | 0.013 |
| T1 | 2 (10.53) | 19 (9.41) | | | | | 9 (2.49) | 142 (4.06) | |
| T2 | 2 (10.53) | 35 (17.33) | | | | | 54 (14.92) | 433 (12.37) | |
| T3 | 2 (10.53) | 21 (10.40) | | | | | 37 (10.22) | 252 (7.20) | |
| T4 | 4 (21.05) | 44 (21.78) | | | | | 53 (14.64) | 733 (20.94) | |
| TX | 4 (21.05) | 17 (8.42) | | | | | 26 (7.18) | 230 (6.57) | |
| Unknown | 5 (26.32) | 66 (32.67) | | 44 (100.00) | 343 (100.00) | | 183 (50.55) | 1711 (48.87) | |
| AJCC N stage, n (%) | | | 0.259 | | | | | | 0.361 |
| N0 | 0 (0.00) | 28 (13.86) | | | | | 25 (6.91) | 325 (9.28) | |
| N1 | 7 (36.84) | 48 (23.76) | | | | | 85 (23.48) | 788 (22.51) | |
| N2 | 1 (5.26) | 21 (10.40) | | | | | 28 (7.73) | 247 (7.06) | |
| N3 | 4 (21.05) | 26 (12.87) | | | | | 30 (8.29) | 257 (7.34) | |
| NX | 2 (10.53) | 13 (6.44) | | | | | 11 (3.04) | 173 (4.94) | |

(*Continued*)

**Table 3.** (Continued)

| Variables | Brain only | | | Other distant sites | | | Multiple Sites | | |
|---|---|---|---|---|---|---|---|---|---|
| | Young patients (n = 19) | Old patients (n = 202) | P | Young patients (n = 44) | Old patients (n = 343) | P | Young patients (n = 362) | Old patients (n = 3501) | P |
| Unknown | 5 (26.32) | 66 (32.67) | | 44 (100.00) | 343 (100.00) | | 183 (50.55) | 1711 (48.87) | |
| Histological type, n (%) | | | 0.554 | | | 0.061 | | | 0.004 |
| IDC | 15 (78.95) | 128 (63.37) | | 41 (93.18) | 262 (76.38) | | 289 (79.83) | 2533 (72.35) | |
| ILC | 0 (0.00) | 11 (5.45) | | 1 (2.27) | 24 (7.00) | | 7 (1.93) | 192 (5.48) | |
| IDLC | 1 (5.26) | 9 (4.46) | | 1 (2.27) | 11 (3.21) | | 10 (2.76) | 101 (2.88) | |
| Other | 3 (15.79) | 54 (26.73) | | 1 (2.27) | 46 (13.41) | | 56 (15.47) | 675 (19.28) | |
| Subtype, n (%) | | | 0.175 | | | 0.050 | | | <0.001 |
| HR+/HER2+ | 1 (5.26) | 31 (15.35) | | 6 (13.64) | 57 (16.62) | | 87 (24.03) | 580 (16.57) | |
| HR+/HER2- | 3 (15.79) | 69 (34.16) | | 12 (27.27) | 162 (47.23) | | 152 (41.99) | 1717 (49.04) | |
| HR-/HER2+ | 3 (15.79) | 25 (12.38) | | 6 (13.64) | 38 (11.08) | | 49 (13.54) | 348 (9.94) | |
| HR-/HER2- | 8 (42.11) | 53 (26.24) | | 16 (36.36) | 70 (20.41) | | 46 (12.71) | 491 (14.02) | |
| Unknown | 4 (21.05) | 24 (11.88) | | 4 (9.09) | 16 (4.66) | | 28 (7.73) | 365 (10.43) | |
| Chemotherapy, n (%) | | | 0.004 | | | 0.003 | | | <0.001 |
| No/unknown | 2 (10.53) | 90 (44.55) | | 1 (2.27) | 73 (21.28) | | 57 (15.75) | 1341 (38.30) | |
| Yes | 17 (89.47) | 112 (55.45) | | 43 (97.73) | 270 (78.72) | | 305 (84.25) | 2160 (61.70) | |
| Surgery, n (%) | | | 0.240 | | | 0.209 | | | <0.001 |
| No | 9 (47.37) | 134 (66.34) | | 4 (9.09) | 66 (19.24) | | 189 (52.21) | 2332 (66.61) | |
| BCS | 2 (10.53) | 21 (10.40) | | 11 (25.00) | 87 (25.36) | | 46 (12.71) | 356 (10.17) | |
| Mastectomy | 8 (42.11) | 45 (22.28) | | 29 (65.91) | 186 (54.23) | | 121 (33.43) | 767 (21.91) | |
| Unknown type | 0 (0.00) | 2 (0.99) | | 0 (0.00) | 4 (1.17) | | 6 (1.66) | 46 (1.31) | |
| Radiotherapy, n (%) | | | 0.746 | | | 0.019 | | | 0.383 |
| No | 2 (10.53) | 31 (15.35) | | 11 (25.00) | 149 (43.44) | | 99 (27.35) | 884 (25.25) | |
| Yes | 17 (89.47) | 171 (84.65) | | 33 (75.00) | 194 (56.56) | | 263 (72.65) | 2617 (74.75) | |
| Combination therapy, n (%) | | | 0.257 | | | 0.016 | | | <0.001 |
| Non-surgery/non-radiotherapy | 0 (0.00) | 6 (2.97) | | 1 (2.27) | 19 (5.54) | | 4 (1.10) | 157 (4.48) | |
| Non-surgery/radiotherapy | 9 (47.37) | 128 (63.37) | | 3 (6.82) | 47 (13.70) | | 185 (51.10) | 2175 (62.13) | |
| Surgery/non-radiotherapy | 2 (10.53) | 25 (12.38) | | 10 (22.73) | 130 (37.90) | | 95 (26.24) | 727 (20.77) | |
| Surgery/radiotherapy | 8 (42.11) | 43 (21.29) | | 30 (68.18) | 147 (42.86) | | 78 (21.55) | 442 (12.62) | |
| Survival, n (%) | | | 0.282 | | | 0.063 | | | <0.001 |
| Yes | 7 (36.84) | 51 (25.25) | | 38 (86.36) | 252 (73.47) | | 158 (43.65) | 1089 (31.11) | |
| No | 12 (63.16) | 151 (74.75) | | 6 (13.64) | 91 (26.53) | | 204 (56.35) | 2412 (68.89) | |
| Died from BC | 12 (100.00) | 144 (95.36) | 0.446 | 6 (100.00) | 85 (93.41) | 0.516 | 192 (94.12) | 2274 (94.28) | 0.924 |
| Died from other diseases | 0 (0.00) | 7 (4.64) | | 0 (0.00) | 6 (6.59) | | 12 (5.88) | 138 (5.72) | |
| Survival time, months, M (Q₁, Q₃) | 10.00 (5.00, 26.00) | 11.00 (3.00, 30.00) | 0.810 | 17.00 (9.00, 30.50) | 15.00 (8.00, 28.00) | 0.514 | 21.00 (10.00, 36.00) | 13.00 (4.00, 30.00) | <0.001 |

Statistics included t test, chi-square test, and rank sum test.

SD: standard deviation, LIQ: lower inner quadrant, LOQ: lower outer quadrant, UIQ: upper inner quadrant, UOQ: upper outer quadrant, AJCC: the American Joint Committee on Cancer stage, IDC: invasive ductal carcinoma, ILC: invasive lobular carcinoma, IDLC: infiltrating ductal mixed lobular carcinoma, HR: hormone receptor, HER: human epidural growth factor receptor, BCS: breast-conserving surgery, BC: breast cancer, M: median, Q1: the 1st quartile, Q3: the 3rd quartile.

patients with bone metastasis who not receive surgery or their surgery type was unknown, and received non-surgery combined with radiotherapy had higher OS and CSS, while those received BCS had lower. In young patients with liver metastasis, surgery combined with radiotherapy was associated with higher OS, while in those who had multiple sites metastasis, not

**Table 4. Association between metastasis patterns and prognosis in young and older patients.**

| Metastatic sites | OS | | | | CSS | | | |
|---|---|---|---|---|---|---|---|---|
| | Model 1 | | Model 2 | | Model 1 | | Model 2 | |
| | HR (95% CI) | P | HR (95% CI) | P | HR (95% CI) | P | HR (95% CI) | P |
| **Young patients** | | | | | | | | |
| Bone only | Ref | | Ref | | Ref | | Ref | |
| Brain only | 3.66 (2.04–6.57) | <0.001 | 2.28 (1.23–4.24) | 0.009 | 3.73 (1.74–8.00) | <0.001 | 2.61 (1.27–5.35) | 0.009 |
| Liver only | 0.80 (0.58–1.12) | 0.193 | 0.96 (0.68–1.37) | 0.835 | 0.80 (0.57–1.11) | 0.174 | 0.93 (0.64–1.36) | 0.719 |
| Lung only | 1.51 (1.09–2.11) | 0.014 | 0.98 (0.68–1.43) | 0.925 | 1.44 (1.01–2.04) | 0.042 | 1.03 (0.68–1.55) | 0.894 |
| Other distant sites | 0.74 (0.33–1.68) | 0.475 | 0.69 (0.30–1.60) | 0.385 | 0.75 (0.33–1.71) | 0.493 | 0.86 (0.40–1.86) | 0.704 |
| Multiple sites | 2.03 (1.67–2.48) | <0.001 | 1.72 (1.39–2.14) | <0.001 | 1.84 (1.51–2.25) | <0.001 | 1.55 (1.24–1.93) | <0.001 |
| **Older patients** | | | | | | | | |
| Bone only | Ref | | Ref | | Ref | | Ref | |
| Brain only | 2.22 (1.88–2.62) | <0.001 | 1.58 (1.33–1.87) | <0.001 | 2.29 (1.89–2.77) | <0.001 | 1.67 (1.34–2.07) | <0.001 |
| Liver only | 1.00 (0.88–1.14) | 0.971 | 1.32 (1.16–1.51) | <0.001 | 1.01 (0.88–1.16) | 0.891 | 1.24 (1.07–1.44) | 0.005 |
| Lung only | 1.29 (1.17–1.43) | <0.001 | 1.10 (0.99–1.22) | 0.089 | 1.24 (1.12–1.38) | <0.001 | 1.07 (0.95–1.21) | 0.247 |
| Other distant sites | 0.83 (0.67–1.03) | 0.085 | 1.03 (0.83–1.28) | 0.780 | 0.84 (0.68–1.03) | 0.100 | 1.11 (0.89–1.37) | 0.347 |
| Multiple sites | 2.01 (1.90–2.13) | <0.001 | 1.86 (1.75–1.98) | <0.001 | 2.02 (1.91–2.15) | <0.001 | 1.88 (1.75–2.01) | <0.001 |

OS: overall survival, CSS: cancer-specific survival, HR: hazard ratio, CI: confidence interval, Ref: reference.

Model 1 was crude model;

Model 2 for young patients adjusted for race, marital status, family income, tumor size, AJCC T and N stage, histological type, subtype, metastatic site, chemotherapy, and surgery;

Model 2 for older patients adjusted for age, race, marital status, family income, primary site, tumor grade, tumor size, AJCC T and N stage, histological type, subtype, metastatic site, chemotherapy, surgery, and radiotherapy.

receive surgery was associated with high risk of lower OS. The details of subgroup analysis results in older patients with BC are showed in Table 6. Generally speaking, not receive radio-therapy, surgery, or combination therapy were positively associated with lower OS and CSS in older patients with whatever the metastasis site was. Additionally, surgery combined with

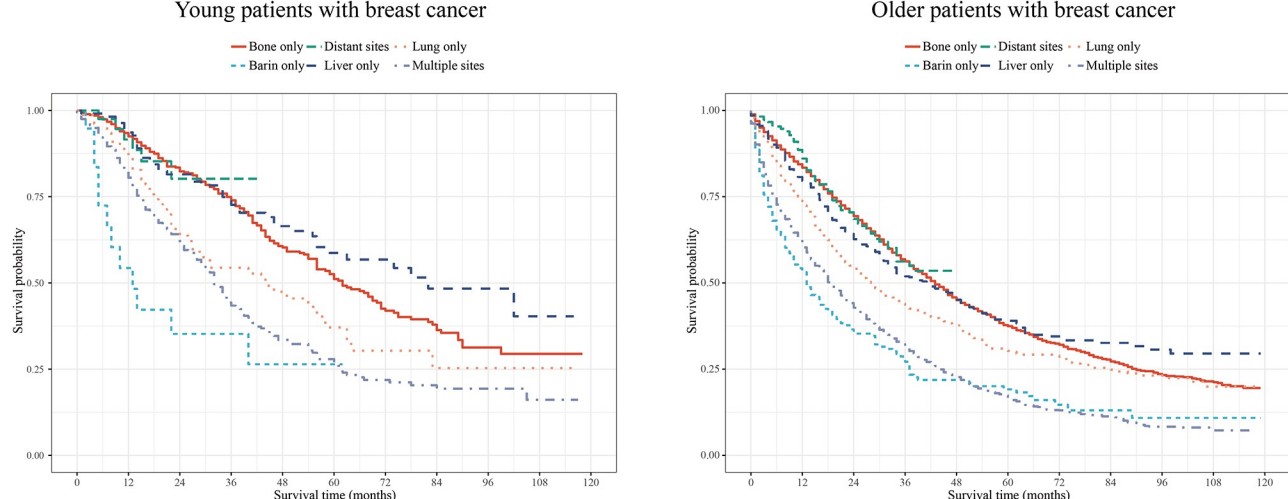

**Fig 2. KM curves of the survival probability in young and older patients with BC who have different metastasis patterns respectively.**

**Table 5. Association between different treatments and prognosis in young and older patients.**

| Variables | OS | | | | CSS | | | |
|---|---|---|---|---|---|---|---|---|
| | Model 1 | | Model 2 | | Model 1 | | Model 2 | |
| | HR (95% CI) | *P* | HR (95% CI) | *P* | HR (95% CI) | *P* | HR (95% CI) | *P* |
| **Young patients** | | | | | | | | |
| Radiotherapy | | | | | | | | |
| Yes | Ref | | Ref | | Ref | | Ref | |
| No | 1.01 (0.84–1.22) | 0.899 | 1.47 (1.16–1.85) | 0.001 | 0.98 (0.81–1.20) | 0.875 | 1.40 (1.10–1.79) | 0.006 |
| Surgery | | | | | | | | |
| Mastectomy | Ref | | Ref | | Ref | | Ref | |
| No | 1.94 (1.60–2.35) | <0.001 | 1.81 (1.44–2.26) | <0.001 | 1.86 (1.54–2.26) | <0.001 | 1.60 (1.26–2.03) | <0.001 |
| BCS | 0.70 (0.53–0.92) | 0.012 | 0.74 (0.55–0.98) | 0.039 | 0.68 (0.51–0.91) | 0.009 | 0.73 (0.53–0.99) | 0.042 |
| Unknown type | 2.25 (1.23–4.13) | 0.009 | 1.89 (0.99–3.60) | 0.055 | 2.35 (1.16–4.77) | 0.018 | 2.13 (1.00–4.53) | 0.051 |
| Combination therapy | | | | | | | | |
| Surgery/non-radiotherapy | Ref | | Ref | | Ref | | Ref | |
| Non-surgery/non-radiotherapy | 1.97 (0.81–4.82) | 0.135 | 1.88 (0.72–4.92) | 0.198 | 2.13 (0.67–6.74) | 0.197 | 2.12 (0.58–7.70) | 0.254 |
| Non-surgery/radiotherapy | 1.64 (1.33–2.03) | <0.001 | 1.55 (1.22–1.98) | <0.001 | 1.63 (1.31–2.04) | <0.001 | 1.42 (1.09–1.84) | 0.009 |
| Surgery/radiotherapy | 0.65 (0.51–0.81) | <0.001 | 0.67 (0.53–0.85) | 0.001 | 0.69 (0.55–0.87) | 0.002 | 0.71 (0.56–0.90) | 0.005 |
| **Older patients** | | | | | | | | |
| Radiotherapy | | | | | | | | |
| Yes | Ref | | Ref | | Ref | | Ref | |
| No | 0.93 (0.88–0.98) | 0.010 | 1.29 (1.20–1.39) | <0.001 | 0.89 (0.84–0.94) | <0.001 | 1.20 (1.11–1.30) | <0.001 |
| Surgery | | | | | | | | |
| Mastectomy | Ref | | Ref | | Ref | | Ref | |
| No | 1.87 (1.76–1.99) | <0.001 | 2.16 (1.99–2.35) | <0.001 | 1.82 (1.71–1.93) | <0.001 | 2.00 (1.84–2.18) | <0.001 |
| BCS | 0.92 (0.84–1.00) | 0.054 | 1.05 (0.96–1.15) | 0.258 | 0.92 (0.84–1.00) | 0.054 | 1.06 (0.96–1.16) | 0.252 |
| Unknown type | 1.30 (1.03–1.63) | 0.026 | 1.18 (0.94–1.49) | 0.161 | 1.29 (1.04–1.61) | 0.022 | 1.21 (0.97–1.50) | 0.088 |
| Combination therapy | | | | | | | | |
| Surgery/non-radiotherapy | Ref | | Ref | | Ref | | Ref | |
| Non-surgery/non-radiotherapy | 3.37 (2.95–3.86) | <0.001 | 2.75 (2.39–3.16) | <0.001 | 2.98 (2.51–3.53) | <0.001 | 2.44 (2.03–2.93) | <0.001 |
| Non-surgery/radiotherapy | 1.58 (1.48–1.68) | <0.001 | 1.66 (1.55–1.78) | <0.001 | 1.59 (1.49–1.70) | <0.001 | 1.65 (1.53–1.78) | <0.001 |
| Surgery/radiotherapy | 0.72 (0.66–0.78) | <0.001 | 0.84 (0.78–0.92) | <0.001 | 0.77 (0.71–0.84) | <0.001 | 0.90 (0.82–0.98) | 0.012 |

OS: overall survival, CSS: cancer-specific survival, HR: hazard ratio, CI: confidence interval, Ref: reference, BCS: breast-conserving surgery.

Model 1 was crude model;

Model 2 for young patients in radiotherapy subgroup adjusted for race, marital status, family income, tumor size, AJCC T and N stage, histological type, subtype, metastatic site, chemotherapy, and surgery;

Model 2 for young patients in surgery and combination therapy subgroups adjusted for race, marital status, family income, tumor size, AJCC T and N stage, histological type, subtype, metastatic site, and chemotherapy;

Model 2 for older patients in radiotherapy subgroup adjusted for age, race, marital status, family income, primary site, tumor grade, tumor size, AJCC T and N stage, histological type, subtype, metastatic site, chemotherapy, and surgery;

Model 2 for older patients in surgery subgroup adjusted for age, race, marital status, family income, primary site, tumor grade, tumor size, AJCC T and N stage, histological type, subtype, metastatic site, chemotherapy, and radiotherapy;

Model 2 for older patients in combination therapy subgroup adjusted for age, race, marital status, family income, primary site, tumor grade, tumor size, AJCC T and N stage, histological type, subtype, metastatic site, and chemotherapy.

**Table 6. Relationship between treatments and prognosis in subgroup of different metastasis patterns.**

| Subgroups | Young patients | | | | Older patients | | | |
|---|---|---|---|---|---|---|---|---|
| | OS | | CSS | | OS | | CSS | |
| | HR (95% CI) | P | HR (95% CI) | P | HR (95% CI) | P | HR (95% CI) | P |
| **Bone only (n = 472)** | | | | | | | | |
| Radiotherapy | | | | | | | | |
| Yes | Ref | | Ref | | Ref | | Ref | |
| No | 0.82 (0.58–1.17) | 0.273 | 0.81 (0.55–1.17) | 0.260 | 1.17 (1.04–1.32) | 0.007 | 1.10 (0.98–1.24) | 0.120 |
| Surgery | | | | | | | | |
| Mastectomy | Ref | | Ref | | Ref | | Ref | |
| No | 2.44 (1.68–3.53) | <0.001 | 2.44 (1.69–3.54) | <0.001 | 2.15 (1.89–2.44) | <0.001 | 1.91 (1.66–2.18) | <0.001 |
| BCS | 0.57 (0.35–0.94) | 0.028 | 0.56 (0.33–0.94) | 0.028 | 0.99 (0.86–1.15) | 0.917 | 1.02 (0.87–1.19) | 0.806 |
| Unknown type | 4.06 (1.47–11.21) | 0.007 | 4.29 (1.31–14.12) | 0.016 | 1.34 (0.91–1.98) | 0.137 | 1.39 (0.97–2.00) | 0.074 |
| Combination therapy | | | | | | | | |
| Surgery/non-radiotherapy | Ref | | Ref | | Ref | | Ref | |
| Non-surgery/non-radiotherapy | 1.90 (0.48–7.55) | 0.362 | 1.97 (0.38–10.16) | 0.420 | 2.92 (2.24–3.79) | <0.001 | 2.45 (1.72–3.47) | <0.001 |
| Non-surgery/radiotherapy | 2.34 (1.57–3.51) | <0.001 | 2.40 (1.57–3.65) | <0.001 | 1.79 (1.59–2.02) | <0.001 | 1.74 (1.53–1.98) | <0.001 |
| Surgery/radiotherapy | 0.75 (0.50–1.13) | 0.170 | 0.77 (0.50–1.18) | 0.226 | 0.89 (0.78–1.01) | 0.066 | 0.97 (0.85–1.10) | 0.611 |
| **Brain only (n = 19)** | | | | | | | | |
| Radiotherapy | | | | | | | | |
| Yes | Ref | | Ref | | Ref | | Ref | |
| No | - | - | - | - | 1.44 (0.80–2.60) | 0.227 | 1.24 (0.67–2.29) | 0.501 |
| Surgery | | | | | | | | |
| Mastectomy | Ref | | Ref | | Ref | | Ref | |
| No | - | | - | | 2.42 (1.44–4.09) | <0.001 | 2.21 (1.35–3.63) | 0.002 |
| BCS | - | | - | | 1.97 (0.93–4.17) | 0.078 | 1.71 (0.84–3.46) | 0.137 |
| Unknown type | - | | - | | 0.34 (0.02–6.79) | 0.479 | 0.37 (0.06–2.20) | 0.276 |
| Combination therapy | | | | | | | | |
| Surgery/non-radiotherapy | Ref | | Ref | | Ref | | Ref | |
| Non-surgery/non-radiotherapy | - | | - | | 3.45 (1.10–10.75) | 0.033 | 4.17 (1.31–13.30) | 0.016 |
| Non-surgery/radiotherapy | - | | - | | 1.46 (0.82–2.61) | 0.199 | 1.82 (0.98–3.40) | 0.059 |
| Surgery/radiotherapy | - | | - | | 0.74 (0.38–1.43) | 0.365 | 1.02 (0.50–2.06) | 0.962 |
| **Liver only (n = 114)** | | | | | | | | |
| Radiotherapy | | | | | | | | |
| Yes | Ref | | Ref | | Ref | | Ref | |
| No | 2.05 (0.92–4.55) | 0.079 | 2.51 (0.93–6.77) | 0.069 | 1.93 (1.42–2.62) | <0.001 | 1.63 (1.23–2.17) | <0.001 |
| Surgery | | | | | | | | |
| Mastectomy | Ref | | Ref | | Ref | | Ref | |
| No | 2.31 (0.42–12.81) | 0.337 | 0.54 (0.02–15.44) | 0.722 | 3.03 (1.89–4.86) | <0.001 | 2.11 (1.22–3.63) | 0.007 |
| BCS | 0.60 (0.18–2.04) | 0.412 | 0.48 (0.12–2.01) | 0.316 | 1.10 (0.81–1.50) | 0.546 | 1.24 (0.90–1.71) | 0.187 |
| Unknown type | 0.89 (0.03–28.24) | 0.949 | 1.21 (0.00–401.57) | 0.948 | 0.96 (0.39–2.39) | 0.931 | 0.72 (0.32–1.62) | 0.430 |
| Combination therapy | | | | | | | | |
| Surgery/non-radiotherapy | Ref | | Ref | | Ref | | Ref | |
| Non-surgery/non-radiotherapy | - | - | - | - | 4.45 (2.02–9.81) | <0.001 | 3.71 (1.38–10.00) | 0.009 |
| Non-surgery/radiotherapy | 1.62 (0.30–8.67) | 0.573 | 0.26 (0.00–13.19) | 0.498 | 1.44 (0.88–2.37) | 0.144 | 1.12 (0.64–1.98) | 0.686 |
| Surgery/radiotherapy | 0.41 (0.17–0.96) | 0.039 | 0.42 (0.16–1.10) | 0.078 | 0.58 (0.42–0.79) | <0.001 | 0.68 (0.50–0.93) | 0.016 |
| Lung only (n = 78) | | | | | | | | |
| Radiotherapy | | | | | | | | |
| Yes | Ref | | Ref | | Ref | | Ref | |

*(Continued)*

**Table 6.** (Continued)

| Subgroups | Young patients | | | | Older patients | | | |
|---|---|---|---|---|---|---|---|---|
| | OS | | CSS | | OS | | CSS | |
| | HR (95% CI) | *P* | HR (95% CI) | *P* | HR (95% CI) | *P* | HR (95% CI) | *P* |
| No | 1.31 (0.55–3.11) | 0.541 | 1.01 (0.38–2.66) | 0.989 | 1.62 (1.28–2.04) | <0.001 | 2.33 (1.43–3.80) | <0.001 |
| Surgery | | | | | | | | |
| Mastectomy | Ref | | Ref | | Ref | | Ref | |
| No | 0.66 (0.15–2.87) | 0.575 | 0.91 (0.23–3.63) | 0.899 | 2.56 (1.92–3.42) | <0.001 | 2.32 (1.69–3.18) | <0.001 |
| BCS | 0.17 (0.03–1.05) | 0.056 | 0.12 (0.00–2.75) | 0.182 | 1.04 (0.80–1.35) | 0.764 | 0.93 (0.71–1.22) | 0.598 |
| Unknown type | - | - | - | - | 1.43 (0.73–2.80) | 0.298 | 1.03 (0.46–2.30) | 0.941 |
| Combination therapy | | | | | | | | |
| Surgery/non-radiotherapy | Ref | | Ref | | Ref | | Ref | |
| Non-surgery/non-radiotherapy | - | - | - | - | 2.01 (1.29–3.13) | 0.002 | 2.12 (1.26–3.58) | 0.005 |
| Non-surgery/radiotherapy | 0.69 (0.17–2.88) | 0.614 | 1.00 (0.24–4.11) | 0.999 | 1.78 (1.36–2.34) | <0.001 | 1.79 (1.31–2.43) | <0.001 |
| Surgery/radiotherapy | 0.80 (0.29–2.16) | 0.655 | 0.99 (0.35–2.80) | 0.984 | 0.60 (0.46–0.78) | <0.001 | 0.69 (0.53–0.90) | 0.006 |
| **Distant sites (n = 44)** | | | | | | | | |
| Radiotherapy | | | | | | | | |
| Yes | Ref | | Ref | | Ref | | Ref | |
| No | - | - | - | - | 2.32 (1.38–3.90) | 0.002 | 1.38 (1.09–1.75) | 0.008 |
| Surgery | | | | | | | | |
| Mastectomy | Ref | | Ref | | Ref | | Ref | |
| No | - | | - | | 1.84 (0.99–3.42) | 0.056 | 2.41 (1.18–4.95) | 0.016 |
| BCS | - | | - | | 0.60 (0.32–1.13) | 0.116 | 0.80 (0.43–1.51) | 0.497 |
| Unknown type | - | | - | | - | - | - | - |
| Combination therapy | | | | | | | | |
| Surgery/non-radiotherapy | Ref | | Ref | | 1.19 (0.62–2.26) | 0.599 | 1.35 (0.64–2.85) | 0.438 |
| Non-surgery/non-radiotherapy | - | | - | | 3.91 (1.63–9.36) | 0.002 | 4.32 (1.81–10.29) | <0.001 |
| Non-surgery/radiotherapy | - | | - | | Ref | | Ref | |
| Surgery/radiotherapy | - | | - | | 0.58 (0.33–1.02) | 0.058 | 0.54 (0.30–0.97) | 0.040 |
| **Multiple sites (n = 362)** | | | | | | | | |
| Radiotherapy | | | | | | | | |
| Yes | Ref | | Ref | | Ref | | Ref | |
| No | 1.03 (0.73–1.45) | 0.869 | 1.01 (0.70–1.44) | 0.976 | 1.23 (1.08–1.39) | 0.001 | 1.17 (1.03–1.33) | 0.018 |
| Surgery | | | | | | | | |
| Mastectomy | Ref | | Ref | | Ref | | Ref | |
| No | 1.63 (1.14–2.35) | 0.008 | 1.30 (0.89–1.90) | 0.179 | 2.05 (1.79–2.35) | <0.001 | 1.96 (1.71–2.25) | <0.001 |
| BCS | 1.09 (0.66–1.82) | 0.734 | 1.00 (0.59–1.68) | 1.000 | 1.11 (0.95–1.31) | 0.200 | 1.11 (0.94–1.31) | 0.209 |
| Unknown type | 1.32 (0.43–4.04) | 0.629 | 1.64 (0.81–3.31) | 0.167 | 1.14 (0.79–1.65) | 0.490 | 1.27 (0.94–1.73) | 0.120 |
| Combination therapy | | | | | | | | |
| Surgery/non-radiotherapy | Ref | | Ref | | Ref | | Ref | |
| Non-surgery/non-radiotherapy | 3.74 (0.84–16.60) | 0.082 | 4.21 (0.65–27.07) | 0.130 | 2.66 (2.16–3.27) | <0.001 | 2.42 (1.89–3.10) | <0.001 |
| Non-surgery/radiotherapy | 1.35 (0.92–1.97) | 0.124 | 1.19 (0.79–1.78) | 0.398 | 1.63 (1.46–1.82) | <0.001 | 1.63 (1.46–1.82) | <0.001 |

(*Continued*)

**Table 6.** (Continued)

| Subgroups | Young patients | | | | Older patients | | | |
|---|---|---|---|---|---|---|---|---|
| | OS | | CSS | | OS | | CSS | |
| | HR (95% CI) | *P* | HR (95% CI) | *P* | HR (95% CI) | *P* | HR (95% CI) | *P* |
| Surgery/radiotherapy | 0.65 (0.42–1.01) | 0.055 | 0.80 (0.52–1.23) | 0.317 | 0.94 (0.81–1.10) | 0.444 | 0.96 (0.83–1.11) | 0.574 |

OS: overall survival, CSS: cancer-specific survival, HR: hazard ratio, CI: confidence interval, Ref: reference, BCS: breast-conserving surgery.

-: Unable to calculate due to sample size.

Model 1 was crude model;

Model 2 for young patients in radiotherapy subgroup adjusted for race, marital status, family income, tumor size, AJCC T and N stage, histological type, subtype, chemotherapy, and surgery;

Model 2 for young patients in surgery and combination therapy subgroups adjusted for race, marital status, family income, tumor size, AJCC T and N stage, histological type, subtype, subtype, and chemotherapy;

Model 2 for older patients in radiotherapy subgroup adjusted for age, race, marital status, family income, primary site, tumor grade, tumor size, AJCC T and N stage, histological type, subtype, chemotherapy, and surgery;

Model 2 for older patients in surgery adjusted for age, race, marital status, family income, primary site, tumor grade, tumor size, AJCC T and N stage, histological type, subtype, metastatic site, chemotherapy, and radiotherapy;

Model 2 for older patients in surgery and combination subgroup adjusted for age, race, marital status, family income, primary site, tumor grade, tumor size, AJCC T and N stage, histological type, subtype, and chemotherapy.

radiotherapy was associated with higher OS and CSS in patients with liver metastasis or lung metastasis.

## Discussion

BC is a major global health problem and a leading cause of death in women [1]. In this study, we compared the characteristics of BC between young patients and older patients, and explored the associations between metastasis patterns and treatment and BC prognosis. The results showed that characteristics, survival time, and metastasis patterns between young and older patients with BC were different. Not receiving radiotherapy or surgery were risk factors for OS and CSS in patients with BC. However, BCS and surgery combined with radiotherapy were associated with higher OS and CSS in young patients, while older patients who had surgery combined with radiotherapy had higher OS and CSS.

The incidence of BC in younger women is rising, and BC patients with different age and molecular subtypes have different metastasis patterns and survival [15]. It is necessary to compare the characteristics metastasis patterns and treatment methods between young and older patients with BC. In a multi-institutional report from China, researchers compared the characteristics and prognosis between very young women and older women with BC, and demonstrated that young patients, compared with older patients, had higher-grade tumors, more probability of lymph vascular invasion in tumor, and more triple-negative subtype [16]. BC patients had less favorable survival outcomes, especially for those with the HR+/HER2- subtype [16]. Another retrospective comparative study from the Madinah region of Saudi Arabia compared clinicopathological characteristics of young versus older patients with BC, and showed that tumor size, tumor grade, tumor stage, lymph vascular invasion, and distant metastasis were significantly different between them [17]. BCs in young patients were triple-negative in most cases and less likely to be ER and progesterone receptor hormone receptors-positive [17]. Compared with older patients, young women more likely exhibited the HER2/neu type and less likely exhibited the tumor type luminal A [17]. Wang et al. [18] used propensity score adjustment method to compare the BCS with mastectomy in young and old women with

early-stage BC, and they did not find any significant difference between BCS and mastectomy in local recurrence, distant disease-free survival, disease-free survival, and breast cancer-specific survival in the young age group, while in the old age group, women who underwent BCS exhibited improved distant disease-free survival. In the current study, we additionally compared the characteristics between young and older patients with different metastasis patterns, including bone only, liver only, lung only, brain only, other distant sites, and multiple sites. The results showed that between young and older patients with bone metastasis, race, marital tatus, primary site, tumor grade, AJCC T and N stage, histological type, subtype, surgery, combination therapy, and survival were all significantly different. Young or older patients who had liver metastasis shared the difference of survival. Race and survival time were different between young and older patients with lung metastasis. Radiotherapy and combination therapy were found to be different between young patients and older patients who had other distant sites metastasis. In addition, variables including race, tumor grade, AJCC T stage, histological type, subtype, chemotherapy, surgery, combination therapy, and survival were all significantly different in patients who had multiple sites metastasis between the two groups.

In our cohort, young patients accounted for 10.68% of the total BC cases, and the mortality in young patients was 45.64%, which was comparable with the previous data from women in the United States [19, 20]. It is significantly lower than that reported in many Asian countries [21, 22]. The possible reason is differences in race backgrounds, genetic susceptibilities, social factors, lifestyles, environmental factors, and economic development levels between young women in developing countries and their counterparts in developed countries. However, data in this realm are lacking. Although we found significant difference in primary site, tumor grade, AJCC T and N stage, histological type, subtype, and metastatic site between young and older patients with BC, the category with the largest proportion is "Unknown" and the specific distinctions are still unclear. Study performed by Zhang et al. [23] showed that young metastatic BC patients had larger tumor size, higher rates of lymph node involvement, and more aggressive BC subtypes. Other studies also demonstrated the aggressiveness of BC in young patients [6, 24]. At present, short metastasis-free interval, visceral involvement and crisis, negative hormone receptor and particularly triple-negative subtype, primary endocrine resistance for luminal subtype, and a number of metastasis sites are recognized as poor prognostic factors [25]. In metastatic BC, the most common metastasis sites include the bone, brain, lung, and liver [26]. We explored the OS and CSS in BC with different metastasis sites in young and older patients, and found that young patients with bone, liver, or multiple sites metastases had higher rate of survival and longer survival time than older patients. The metastasis patterns in different age groups remain controversial. A study based on the Epidemiological Strategy and Medical Economics (ESME) database found that young patients with metastatic BC were more likely to have visceral metastasis than bone metastasis [27]. Chen et al. [10] showed that patients with metastatic BC who aged <50 years old were more likely to have distant lymph node metastasis and multiple sites metastases, and less likely to have lung metastasis. Another study showed that young patients with metastatic BC were more likely to have brain and liver metastases than older patients [28]. In our study, older patients with liver metastasis seemed to have higher risk of both lower OS and CSS compared with young patients. Liver metastasis is associated with HER2+, and according to the previous studies, HER2+ and triple-negative subtype are more likely to have visceral metastasis than bone metastasis [29–31]. However, the underlying molecular mechanisms are still needed further explorations.

Moreover, we assessed the associations between different metastasis patterns and BC prognoses in young and older patients who received different treatments. Conventional treatments for BC include radiotherapy, surgery (mastectomy and BCS), and chemotherapy [32]. Our results indicated that beside of surgery combined with radiotherapy, in young

women, BCS seemed to have a good curative effect, which had higher OS and CSS, however, these relationships were not found in older patients. It is inconsistent with the previous data from Wang's study [18], which found that older patients who underwent BCS exhibited improved distant disease-free survival. Actually, younger patients were more likely to receive treatment (surgery and chemotherapy) [10]. In addition to conventional systemic treatment, patients with metastatic BC also benefit from local treatment [33]. The subgroup analyses also showed that radiotherapy combined with surgery may be the first choice for older patients with different metastases, while BCS may be more suitable for young patients with bone metastasis. Possible reason for this discrepancy was undertreatment in elderly patients to some extent. The improvement of OS in patients with metastatic BC is mainly driven by the HER2+ subgroup [34]. In 2013, a new HER2-targeted therapies were released which were associated with major OS benefits in clinical trials [35, 36]. A study on patients with HER2+ metastatic BC found that young patients were more likely to receive PH+taxane than older patients, and older patients were more likely to receive regimens with H without P or hormone therapy, and the results showed that young patients have better CSS than older patients [37]. An observational study showed that chemotherapy and anti-HER2 therapies were less frequently used in older patients, resulting in shorter OS in women who aged >60 years old [38]. Moreover, other treatments for metastatic BC, such as immune-based combinations, are also recognized as promising means of improving BC survival [39, 40]. Nevertheless, in clinical, whether to use different treatment regimens for BC patients of different ages needs further verification.

This study explored the characteristics, metastasis patterns, and treatment methods of young patients with BC, and compared them with older patients, which may provide some references for the identification of high-risk population with metastatic BC in different ages and the accurate formulation of intervention strategies. However, there were still some limitations in our study. Data in the current study were extracted from the SEER database, and patients without the information of treatment were excluded, which may cause the selection bias. Variables such as metastasis sites were collected since 2010, so the overall sample was relatively small, and thus some subgroups could not output results due to the limited sample size. The physical function status and possible disease progression during follow-up of patients were not available, which may influence the results. Also, we only included three common treatments (surgery, radiotherapy, and combination surgery) for BC, other treatments such as chemotherapy and immunotherapy are needed further exploration. Additionally, further study on the choice of treatment in BC patients with different ages and metastasis patterns should be investigated.

## Conclusion

Young patients with BC compared with older patients had different disease characteristics. In clinical, different metastasis sites and treatments in young and older patients may associated with BC prognoses.

## Supporting information

**S1 Table. Covariates related to OS and CSS in young patients with BC.**
(DOCX)

**S2 Table. Covariates related to OS and CSS in older patients with BC.**
(DOCX)

## Author Contributions

**Conceptualization:** Xiaokang Gao, Jie Bian.

**Data curation:** Fengxia Zhang, Qiwang Zhou, Hui Xu.

**Formal analysis:** Fengxia Zhang, Qiwang Zhou, Hui Xu.

**Investigation:** Fengxia Zhang, Qiwang Zhou, Hui Xu.

**Methodology:** Fengxia Zhang, Qiwang Zhou, Hui Xu.

**Writing – original draft:** Xiaokang Gao.

**Writing – review & editing:** Xiaokang Gao, Jie Bian.

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
