## [Decision Letter · Decision Letter 0]

15 Sep 2023

PONE-D-23-27638Metastasis, characteristic, and treatment of breast cancer in young women and older women: A study from the Surveillance, Epidemiology, and End Results registration databasePLOS ONE

Dear Dr. Bian,

Thank you for submitting your manuscript to PLOS ONE. After careful consideration, we feel that it has merit but does not fully meet PLOS ONE’s publication criteria as it currently stands. Therefore, we invite you to submit a revised version of the manuscript that addresses the points raised during the review process.

We look forward to receiving your revised manuscript.

Kind regards,

Alessandro Rizzo

Academic Editor

PLOS ONE

Journal Requirements:

Reviewers' comments:

Reviewer's Responses to Questions

**Comments to the Author**

1. Is the manuscript technically sound, and do the data support the conclusions?

Reviewer #1: Partly

Reviewer #2: Yes

2. Has the statistical analysis been performed appropriately and rigorously? 

Reviewer #1: Yes

Reviewer #2: I Don't Know

3. Have the authors made all data underlying the findings in their manuscript fully available?

Reviewer #1: Yes

Reviewer #2: Yes

4. Is the manuscript presented in an intelligible fashion and written in standard English?

Reviewer #1: Yes

Reviewer #2: No

5. Review Comments to the Author

Reviewer #1: The study assesses a current, timely topic in breast cancer.

We recommend some changes:

- We believe this article is suitable for publication in the journal although major revisions are needed. The main strengths of this paper are that it addresses an interesting and very timely question and provides a clear answer, with some limitations. Certainly, the authors should better highlight the limitations of the current paper. In addition, the samples are quite old.

- A linguistic revision is needed.

- The background of the changing scenario of medical treatment in breast cancer should be better discussed, and some recent papers regarding this topic should be included ( PMID: 35171746 ; PMID: 36633661; PMID: 34802383; PMID: 34793275 ).

Major changes are necessary.

Reviewer #2: Number 3. Datasets - It is concerning that, although the authors say all the datasets are available from SEER, the ones authors generated and/or analyzed are not SEER datasets, but their analysis. Are they available?

- The authors do not define "younger" and "older" patients by age until into the Introduction. This should be done in the Abstract. Also, OS and CSS are not defined, although one can presume they mean Overall Survival and Cancer Specific Survival. In the Results section, it is stated that IDC was the most common histological type; one assumes the authors mean Invasive Ductal Carcinoma, which also should be spelled out.

Number 4 - There are problems with the language used by the authors. It is usually understandable in the sense that one can figure out what the authors mean, but they need to have a native English speaker correct many odd phrases. There are repeated instances such as "died for" instead of "died from", "besides" instead of "in addition", "found differently" instead of "found to be different," and most annoying, "high risk of OS and CSS". The patients are not at high risk of overall survival, but at high risk of lower overall survival (more likely to die) and cancer specific survival.

- In the Methods section, the authors state the database used from SEER was from 2000-2019, but in Figure 1, it is 2010-2019.

There is confusion as to what HER2 means; in Variables collection it states "...and hormonal estrogen receptor (HER)," and this is repeated in the Discussion. HER stands for Human Epidural Growth Factor receptor, not hormone estrogen receptor; this is an egregious error.

6. PLOS authors have the option to publish the peer review history of their article (what does this mean?). If published, this will include your full peer review and any attached files.

Reviewer #1: No

Reviewer #2: No

---

## [Author Response · Author response to Decision Letter 0]

17 Oct 2023

Responses to Reviewer #1’s comments

The study assesses a current, timely topic in breast cancer.

We recommend some changes:

- We believe this article is suitable for publication in the journal although major revisions are needed. The main strengths of this paper are that it addresses an interesting and very timely question and provides a clear answer, with some limitations. Certainly, the authors should better highlight the limitations of the current paper. In addition, the samples are quite old.

Response: Thank you for your approbation and valuable comments. We have revised this manuscript to better highlight the limitations of the current study. The study sample were extracted from the SEER database, which includes the large numbers of representative population in the United States. SEER uses data collected and maintained by the National Center for Health Statistics (NCHS), which is from the Population Estimates Program (PEP) data of the United States Census Bureau and mortality data in the United States. SEER database routinely collects and publishes data on patient-specific and tumor-specific characteristics for about 30 percent of the US population. We are very sorry for the clerical error that the actual data collection time was 2010-2019. Due to the information of patients such as metastatic sites were collected from 2010, and the SEER database has not public data from 2020 and beyond, wo could only used the data from 2010 to 2019 when this study started.

- A linguistic revision is needed.

Response: Thank you for your comment. We have checked and revised the grammatical mistakes in the manuscript, and invited a native English speaker to help us. We hope the revised manuscript could meet your standards.

- The background of the changing scenario of medical treatment in breast cancer should be better discussed, and some recent papers regarding this topic should be included (PMID: 35171746; PMID: 36633661; PMID: 34802383; PMID: 34793275).

Major changes are necessary.

Response: Thank you for your valuable comment. We have read the references you provided, and added them into the discussion section.

Responses to Reviewer #2’s comments

Number 3. Datasets - It is concerning that, although the authors say all the datasets are available from SEER, the ones authors generated and/or analyzed are not SEER datasets, but their analysis. Are they available?

Response: Thank you for your comment. This was a retrospective cohort study, where the information of participants was extracted from the SEER database, which routinely collects and publishes data on patient-specific and tumor-specific characteristics for about 30 percent of population in the United States. The SEER database was publicly available, and the information of patients was all de-identified. 

- The authors do not define "younger" and "older" patients by age until into the Introduction. This should be done in the Abstract. Also, OS and CSS are not defined, although one can presume, they mean Overall Survival and Cancer Specific Survival. In the Results section, it is stated that IDC was the most common histological type; one assumes the authors mean Invasive Ductal Carcinoma, which also should be spelled out.

Response: Thank you for your valuable comments. We have revised the abstract section and added the definition of “younger” and “older” patients by age as well as that of OS and CSS. We have revised the “infiltrating ductal carcinoma” into “invasive ductal carcinoma”.

Number 4 - There are problems with the language used by the authors. It is usually understandable in the sense that one can figure out what the authors mean, but they need to have a native English speaker correct many odd phrases. There are repeated instances such as "died for" instead of "died from", "besides" instead of "in addition", "found differently" instead of "found to be different," and most annoying, "high risk of OS and CSS". The patients are not at high risk of overall survival, but at high risk of lower overall survival (more likely to die) and cancer specific survival.

Response: Thank you for your valuable comments. We have checked and revised the grammatical mistakes in the manuscript, and invited a native English speaker to help us. We hope the revised manuscript could meet your standards.

- In the Methods section, the authors state the database used from SEER was from 2000-2019, but in Figure 1, it is 2010-2019.

Response: Thank you for your comments. We are very sorry for the clerical error, and have revised the time of data collection we used, that the correct time was 2010-2019.

There is confusion as to what HER2 means; in Variables collection it states "...and hormonal estrogen receptor (HER)," and this is repeated in the Discussion. HER stands for Human Epidural Growth Factor receptor, not hormone estrogen receptor; this is an egregious error.

Response: Thank you for your comments. We are very sorry for this egregious error, and have revised the “hormonal estrogen receptor” into “human epidural growth factor receptor” in the manuscript.

---

## [Editor Report · Decision Letter 1]

19 Oct 2023

Metastasis, characteristic, and treatment of breast cancer in young women and older women: A study from the Surveillance, Epidemiology, and End Results registration database

PONE-D-23-27638R1

Dear Dr. Bian,

We’re pleased to inform you that your manuscript has been judged scientifically suitable for publication and will be formally accepted for publication once it meets all outstanding technical requirements.

Kind regards,

Alessandro Rizzo

Academic Editor

PLOS ONE
---

## [Editor Report · Acceptance letter]

25 Oct 2023

PONE-D-23-27638R1 

Metastasis, characteristic, and treatment of breast cancer in young women and older women: A study from the Surveillance, Epidemiology, and End Results registration database 

Dear Dr. Bian:

I'm pleased to inform you that your manuscript has been deemed suitable for publication in PLOS ONE. Congratulations! Your manuscript is now with our production department. 

Kind regards, 

on behalf of

Dr. Alessandro Rizzo 

Academic Editor

PLOS ONE